# Automated Electrodermal Activity and Facial Expression Analysis for Continuous Pain Intensity Monitoring on the X-ITE Pain Database

**DOI:** 10.3390/life13091828

**Published:** 2023-08-29

**Authors:** Ehsan Othman, Philipp Werner, Frerk Saxen, Ayoub Al-Hamadi, Sascha Gruss, Steffen Walter

**Affiliations:** 1Department of Neuro-Information Technology, Institute for Information Technology and Communications, Otto-von-Guericke University Magdeburg, 39106 Magdeburg, Germany; philipp.werner@ovgu.de (P.W.); frerk.saxen@ovgu.de (F.S.); ayoub.al-hamadi@ovgu.de (A.A.-H.); 2Department of Medical Psychology, Ulm University, 89081 Ulm, Germany; sascha.gruss@uni-ulm.de (S.G.); steffen.walter@uni-ulm.de (S.W.)

**Keywords:** continuous pain intensity recognition, electrodermal activity, facial expressions, fusion, long-short term memory network, random forest, sample weighting

## Abstract

This study focuses on improving healthcare quality by introducing an automated system that continuously monitors patient pain intensity. The system analyzes the Electrodermal Activity (EDA) sensor modality modality, compares the results obtained from both EDA and facial expressions modalities, and late fuses EDA and facial expressions modalities. This work extends our previous studies of pain intensity monitoring via an expanded analysis of the two informative methods. The EDA sensor modality and facial expression analysis play a prominent role in pain recognition; the extracted features reflect the patient’s responses to different pain levels. Three different approaches were applied: Random Forest (RF) baseline methods, Long-Short Term Memory Network (LSTM), and LSTM with the sample-weighting method (LSTM-SW). Evaluation metrics included Micro average F1-score for classification and Mean Squared Error (MSE) and intraclass correlation coefficient (ICC [3, 1]) for both classification and regression. The results highlight the effectiveness of late fusion for EDA and facial expressions, particularly in almost balanced datasets (Micro average F1-score around 61%, ICC about 0.35). EDA regression models, particularly LSTM and LSTM-SW, showed superiority in imbalanced datasets and outperformed guessing (where the majority of votes indicate no pain) and baseline methods (RF indicates Random Forest classifier (RFc) and Random Forest regression (RFr)). In conclusion, by integrating both modalities or utilizing EDA, they can provide medical centers with reliable and valuable insights into patients’ pain experiences and responses.

## 1. Introduction

A reliable assessment of pain is crucial to determine proper and prompt treatment, especially for vulnerable patients who cannot communicate their pain, such as those in intensive care, people with dementia, or adults with cognitive impairment. To make the clinical observations go well, it is promising to provide an automated system due to its possibility for objective and robust measurements and the monitoring of pain [1]. The COVID-19 pandemic has further highlighted the importance of such systems. Many countries like China adopted automated systems to effectively manage patients [2]. Thus, this study aims to develop an automated system for clinical settings that can rapidly and objectively monitor patients’ pain levels by analyzing the informative modalities in the X-ITE Pain Dataset. Such a database has been made to complement existing databases and provide valuable information for more advanced discriminating pain or pain intensities versus no pain.

Physical expressions of pain encompass visual cues (facial expressions and body movements), vocalization cues (verbally and non-verbally), and physiological cues (electrocardiography (ECG), electromyography (EMG), Electrodermal Activity (EDA), and brain activity) [3,4,5]; these cues play a significant role in assessing pain in individuals [5]. The extracted features from EDA and facial expression modalities indicate the spontaneous pain expression, stress, and anxiety caused by different pain levels; both modalities are good measures for pain assessment [6,7]. This study presents the findings obtained from analyzing two important modalities regarding classification and regression. Regarding regression approaches, the pain intensity stimuli were handled as continuous labels and normalized between 0 and 1 to fit using all regression approaches.

EDA records the changes in the skin’s electrical activity using two electrodes attached to the index and ring fingers. It correlates significantly with pain intensity ratings, as it reflects the intense body reactions after experiencing pain when a painful stimulus is applied [8,9,10]. An increasing number of studies [11,12,13,14,15] explored physiological signals and machine learning models for objective assessments of pain intensity; findings demonstrated that EDA signals tend to outperform other physiological signals in terms of accurate pain assessment. Thus, many studies [16,17,18] focused on EDA for pain assessment. Further, the temporal integration of EDA features were investigated to improve the performance of pain assessment [14,19,20]. The temporal integration was represented as a time series statistics descriptor (EDA-D) that was calculated from several statistical measures along with their first and second derivatives per time series.

Ekman and Friesen [21] decomposed facial expressions into individual facial Action Units (AUs) with the Facial Action Coding System (FACS). A combination of some of these AUs expresses pain behaviors [22]. Prior studies [13,14,23,24,25,26,27] using facial expressions have explored machine learning approaches to recognize pain intensity. Regarding the use of the temporal integration of frame-level features represented by the Facial Activity Descriptor (FAD), RF showed superior performance compared to linear Support Vector Machine (SVM) and Radial Basis Function kernel (RBF-SVM) [24]; thus, it was used in [27] and this study as baseline approach regarding classification and regression. Approaches that use FAD to recognize pain intensity showed better results than those that used facial features [24], which relied on independently extracted facial features from each frame of a given sequence. FAD is good at describing dynamics among neighboring frames.

Machines are much better than human observers at recognizing pain intensity via facial expression [26]. As mentioned earlier, the X-ITE Pain Database was used in this work, Walter et al. [20], Werner et al. [14], and Gruss et al. [28] used it as a recent and valuable database. These studies reported the results of using phasic (short) and tonic (long) stimulation samples from a frontal RGB camera, as well as audio and psychological data (ECG, EMG, and EDA) of 7 s, which have been cut out from the continuous recording of the main stimulation phase in the X-ITE Pain Database. Our work in [27] goes beyond their study. We reported the first continuous pain monitoring results based on analyzing facial expressions on the same database. We used the continuous recording of the facial expressions of most of the experiment, which is about 1 h and a half per subject. This study extends our work in [27] by investigating the same proposed methods but with EDA modality regarding classification and regression, then comparing the results with the facial expressions modality results.

In this study, we applied three distinct approaches, utilizing 11 proposed datasets from the X-ITE Pain Database. The objective was to discriminate between no pain and various pain intensities scenarios regarding sequence classification and regression. These scenarios encompassed scenarios involving “no pain” (indicating the absence of painful stimuli), three levels of pain intensity (low, moderate, and severe), as well as variations in pain qualities (heat and electrical stimuli) for two types of pain stimuli (phasic and tonic). Those methods are Random Forest (RF) as baseline methods (Random Forest classifier (RFc) and Random Forest regression (RFr)), Long Short-Term Memory (LSTM), and LSTM using the sample-weighting method (called LSTM-SW). The LSTM [29] was used because it is good for handling time series prediction to improve the continuous monitoring of pain intensity. Both EDA-D and FAD were represented by summarizing four statistics (minimum, maximum, mean, and standard deviation) of the time series itself and its first and second derivatives. The EDA-D we used was at the same frame rate (1/25 s = 25 fps) as that used with FAD in [27]; we used 25 fps because it is the most common and standard frame rate of the video. Further, we used a sliding window of ten seconds ago with the label of three seconds after.

In addition, recent studies [6,13,14,15,30] have motivated us to combine the EDA and facial expressions modalities to improve continuous pain intensity monitoring. The combination of behavioral and physiological pain responses holds the potential for developing objective pain assessment, as discussed by Odhner et al. [31]. The late fusion method, known as Decision Fusion (DF), employs a fixed mapping (mean-score mapping), wherein individual RF, LSTM, and LSTM-SW models are applied to two modalities (FAD and EDA). This study compares the results of the individual modalities (EDA and facial expressions) and the fused modality of these two approaches, aiming to introduce the optimal automated system for the objective and continuous monitoring of pain intensity. Such a system could offer significant benefits for reliable and cost-effective pain intensity assessment.

## 2. Related Work

Several studies of pain have focused on physiological signals because of the strong correlation between these signals and pain [32,33,34]. In [5,14,15], it was reported that the EDA signal obtained the best performance compared to other single physiological signals. Thus, EDA has gained attention in automatic pain recognition systems. EDA records changes in the electrical activity of the skin of the hands, which is controlled by the autonomic nervous system [35,36]. The sweat on the skin’s surface changes the electrical conductivity of the skin (e.g., people sweat when they are scared, nervous, and in pain). EDA is composed of phasic and tonic signals. The phasic signal is a quick response caused by external stimuli such as pain stimuli. The tonic signal is a slower component of the signal, including the baseline of the signal due to unconscious activities [37].

Recent studies have focused on deep-learning methods due to their success in classifying pain using EDA, such as 1D convolutional neural networks [CNNs] [13], a multi-task learning method based on neural networks [38], and the Recurrent Convolutional Neural Network [RCNN] [12]. These deep-learning methods were utilized because of their ability to mine the sequential relationships between different periods of EDA signals. Posada et al. [17] presented classification and regression machine learning models to estimate pain sensation in healthy subjects using EDA. They computed the extracted features of EDA based on time-domain decomposition, spectral analysis, and differential features. The maximum macro-averaged geometric mean scores of the models were 69.7% and 69.2%, respectively. Kong et al. [18] analyzed the spectral characteristics of EDA to obtain reliable performance because it is more sensitive and reproducible for the assessment of sympathetic arousal than traditional indices (tonic and phasic signals). Bhatkar et al. [16] reported a successful novel method to discriminate the reduction in pain with clinically effective analgesics by combining self-reports with continuous physiological data in a structured and specific-to-pain protocol.

A common knowledge is that pain databases have a significant impact on the performance of automatic pain assessment systems. The above-mentioned studies of EDA signals for pain intensity recognition used databases that include fewer variants of quality and duration. By analyzing pain in terms of quality and length, additional valuable information is provided for more advanced discrimination between pain or pain intensity versus no pain. Thus, the X-ITE Pain Database [28] is designed to complement existing databases. The X-ITE Pain Database includes behavioral and physiological data that were recorded when healthy participants (subjects) were exposed to different qualities and durations of pain stimuli. The use of healthy subjects in a medical study has always played a vital role in evaluating safety and tolerability without interference from concomitant pathological conditions [39].

Werner et al. [24] introduced a novel feature set for describing facial actions and their dynamics, which we call facial activity descriptors [FAD]. They trained FAD (extracted from the BioVid Heat Pain Dataset) with SVM and RFc, and the results showed that RFc with 100 trees outperformed SVM. They focused on the video-level using temporal integration for pain recognition because it was more effective in describing the dynamic information beneficial for pain intensity recognition [23]. This approach often involves the temporal integration of frame-level features. For example, video content can be condensed to high-level features using a time series statistics descriptor that consists of several statistical measures of the time series. In [14], the same RFc was trained using the extracted features from facial expressions, audio, ECG, EMG, and EDA that were introduced in [28] to recognize pain levels. They classified the pre-segmented time windows (7 s) cut out from the continuous recording of the main stimulation phase in the X-ITE Pain Database. According to the ability of Random Forest (RF) [40] for pain detection using facial expressions [14,23], we introduced RFc using temporal information of facial expressions by representing time-series statistics descriptor (FAD) [25,26]. FAD was represented by calculating several statistical measures with their first and second derivatives per time series. The performances of reduced MobileNetV2 and simple Convolutional Neural Network (CNN) were better than RFc. CNN accuracy improved when using the sample weighting method by about 1%. The sample weighting method was suggested to reduce the weight of misclassified samples by duplicating some training samples with more facial responses if their classification scores are above 0.3 to improve the pain intensity recognition performance [26].

In [5,6,13,14,20,23,28,30,38], the authors reported that fusing modalities could improve the results of pain recognition. After investigating these studies, it was found that some fused physiological modalities, while others fused both behavioral and physiological modalities. The models combining the fused modalities of EMG and EDA were the most successful in the majority of the aforementioned studies for developing pain recognition systems. However, physiological signals could also be indicative of other pathological conditions unrelated to pain. In the study by Werner et al. [14], fusion was applied with multiple modalities (frontal RGB camera, audio, ECG, EMG, and EDA). Firstly, they individually trained random forests (RF) using the features of each modality. Secondly, they concatenated the feature vectors of all modalities and trained and tested the RF (referred to as feature fusion). Thirdly, they applied decision fusion by training the RF on individual modalities and then aggregating the RF scores into final decisions. They employed two types of aggregation: fixed mapping and trained mapping approaches.

In this study, we recognized pain intensity by utilizing Long Short-Term Memory (LSTM) [29]. LSTM was designed to learn long-term dependencies over extended time periods by retaining information from previous segments. LSTM, when applied with sample weighting (LSTM-SW), exhibited significantly superior performance compared to RFr for recognizing continuous pain intensity when employing facial expressions for regression [27]. Furthermore, the same methods were applied for classification in [41]. The exceptions were the small datasets, for which RFr was the best, but the performance of the models was still poor. Additionally, the results obtained from the facial expressions modality were juxtaposed with the EDA modality results. In alignment with [14], this study uses late fusion with the fused mapping method on two modalities (EDA and facial expressions), a technique referred to as Decision Fusion (DF). We suggest fusing facial expressions (the most informative behavioral modality) and EDA (the most informative physiological modality). We believe that utilizing a combination of behavioral and physiological modalities with appropriate machine learning methods could enhance the performance of pain intensity recognition.

In contrast to [14,14], we used the most continuous recording of facial expressions and EDA signals for classification and regression tasks in this study. This study introduces RFc and RFr as baseline methods for continuously monitoring pain intensity using the X-ITE Pain Database. In our recent study [42], we presented the results of investigations involving multiple modalities (frontal RGB camera, audio, ECG, EMG, and EDA). This work represents a further investigation into analyzing only the two most informative modalities (facial expression and EDA signal), which are then fused using Decision Fusion (DF) on both balanced and imbalanced datasets.

## 3. Materials and Methods

This section introduces the structure of our system designed to automatically recognize continuous pain intensity using the EDA and facial expressions modalities from the X-ITE Pain Database (see Figure 1). The primary focus of this study is on the EDA and the fused modality (EDA and facial expressions) to recognize continuous pain intensity. The analysis of facial expressions was previously described in detail in [27,41]. Firstly, we determined the temporal integration of the extracted features from the EDA and facial expressions modalities, referred to as EDA descriptor (EDA-D) and Facial Activity Descriptor (FAD), respectively. Secondly, we shifted the labels three seconds forward and applied a sliding window with a time length of ten seconds. Thirdly, we used Random Forest (RF) as the baseline method, along with two Long Short-Term Memory (LSTM) methods (one using the sample weighting method and the other without) for recognizing continuous pain intensity. Fourthly, we applied the late fusion method (Decision Fusion (DF)), in which individual RF, LSTM, and LSTM-SW models were trained with EDA-D and FAD. Finally, we evaluated the performance of the models in terms of classification using the Micro average F1-score (Micro avg. F1-score), which is particularly useful when datasets vary in size. Furthermore, the Mean Squared Error (MSE) and the intraclass correlation coefficient (ICC (3, 1)) [43] were computed to compare the performances of classification versus regression models after normalizing the output to a range between 0 and 1. When dealing with continuous data, MSE is a common measure used, and ICC assesses interrater reliability by determining the correlation between two measurements conducted on the same subject.

### 3.1. Database Prepossessing

In this section, we provide an overview of the multimodal Experimentally Induced Thermal and Electrical (X-ITE) Pain Database [28], which we used to validate the performance of various automatic methods for continuous pain intensity recognition. Within this database, only a subset of 127 participants (aged between 18 and 50 years) has data available from all sensors (frontal RGB camera, audio, ECG: electrocardiogram, EMG: surface electromyography, EDA: electrodermal activity). Alongside Werner et al. [14] and our studies [26,27,41], we focused on this subset and analyzed the EDA and facial expressions data from time series involving both phasic and tonic pain intensities at three levels (low, medium, and high). These pain intensities were experienced during the application of thermal pain stimuli (Medoc PATHWAY Model ATS) and electrical pain stimuli (Digitimer DS7A), as well as during periods of no pain. The 5 s phasic stimuli of each modality (heat and electrical pain) and intensity were repeated 30 times in randomized order with pauses of 8–12 s. The tonic stimuli were applied once for one minute per intensity and modality, followed by a pause of five minutes. For further details regarding the data collection experiment, refer to the work of Gruss et al. [28]

Automatic methods should be capable of recognizing pain intensity from facial expressions in frontal RGB videos and EDA data time series. However, we observed that the distribution of samples for pain intensity labels is extremely unbalanced. To address this issue, undersampling, clustering, and oversampling techniques were used to overcome such problem [44]. In alignment with our recent studies [27,41], we addressed the problem of imbalanced data using similar processing steps but with EDA data. We used the results of investigating the intensity of facial expressions for most samples when expressing pain intensity. Firstly, we utilized the same four categories of subjects, categorized according to their expression of pain intensity. Secondly, we used the same splits of the database, which were 80% of data for training (100 subjects = 572,696 samples), 10% for validation (13 subjects = 75,537 samples), and 10% for testing (14 subjects = 79,485 samples); each split contains samples from all intensity categories. The subjects were selected randomly from each category based on the proposed percentage, Figure 2 shows the subjects in each category. Thirdly, we processed the database into 11 datasets based on two pain stimulus types (phasic and tonic) and two qualities (heat and electrical stimuli) to reduce the impact of the imbalanced database problem. The distribution of samples for pain intensity labels within each dataset is detailed in Table 1. The reason why we do not use all samples with the testing set (without splitting it into 11 datasets) is that the model would be biased towards the majority and fail to recognize pain intensity in samples of minority classes. The size of the tonic samples is very small compared to that of the phasic samples. Additionally, the size of the heat samples size is smaller than that of electrical samples for both pain qualities (phasic and tonic).

Table 1 presents the 11 suggested datasets, which are as follows:Phasic Dataset (PD): Excludes tonic samples and no-pain samples both before and after the samples, including those labeled −10 and −11 (indicating experimental issues like false starts, restarts of stimuli, overlap between heat or electrical stimulation, unbalanced phasic estimation, short pauses, short tonic electrical stimulus, single heat stimulus in front, additional stimulus, or subject interaction during the experiment).Heat Phasic Dataset (HPD): Excludes electrical samples from PD and no-pain samples before these frames.Electrical Phasic Dataset (EPD): Excludes heat samples from PD and no-pain frames before these frames.Tonic Dataset (TD): Excludes phasic samples and no-pain samples both before and after the samples, including those labeled −10 and −11.Heat Tonic Dataset (HTD): Excludes electrical samples from TD and no-pain frames before these frames.Electrical Tonic Dataset (ETD): Excludes heat samples from TD and no-pain frames before these frames.

Additionally, the Reduced Subsets are:Reduced Phasic Dataset (RPD): Reduces the no-pain frames in PD to approximately 50%.Reduced Heat Phasic Dataset (RHPD): Reduces the no-pain frames in HPD to approximately 50%.Reduced Electrical Phasic Dataset (REPD): Reduces the no-pain frames in EPD to approximately 50%.Reduced Tonic Dataset (RTD): Reduces the no-pain frames in TD to approximately 38%.Reduced Electrical Tonic Dataset (RETD): Reduces the no-pain frames in ETD to approximately 49%.

Our reduction strategy focuses on minimizing some no-pain samples preceding each pain intensity sequence while retaining varying numbers of no-pain samples immediately adjacent to each pain intensity sequence. This number is determined based on the sample count within each pain intensity sequence. For instance, for a sequence of phasic electrical pain intensity that comprises five samples, we retain the preceding five no-pain samples and discard the remainder.

### 3.2. Processing of Electrodermal Activity (EDA) and Facial Expressions Data

After preprocesssing database (see the section above), we processed the data of both modalities (EDA and facial expressions) to extract features for continuous recognising pain intensity. In line with the facial expressions analysis process in our study [27], we used only the EDA signal (without filtering) at the same time series sampling rate (1/25 s). The Facial Features (FF) were extracted from each frame for each video (subject) using OpenFace [45]; the average length of videos was about one and a half hours. OpenFace detected the face and facial landmarks, extracted Action Units (AUs), and estimated head pose. The FF were recorded at 25 frames per second (fps). The FF we used include 21 features: 3 head pose (Yaw, Pitch, and Roll), AU1 (binary occurrence output), and 17 AU intensity outputs of OpenFace, which are AU1, AU2, AU4, AU5, AU6, AU7, AU9, AU10, AU12, AU14, AU15, AU17, AU20, AU23, AU25, AU26, and AU45. Temporal integration features were computed from the 1-dimensional EDA time series and the 21-dimensional facial expression time series. Each time series was summarized using four statistics derived from the time series itself and its first and second derivatives: minimum, maximum, mean, and standard deviation. The obtained descriptors (Facial Activity Descriptor (FAD) and EDA Descriptor (EDA-D)) yield a 12 × 1-dimensional and 12 × 21-dimensional descriptor per time series for EDA and FF features, respectively. A person-specific standardization of the features [24] was applied with both descriptors in order to focus on the within-subject response variation rather than the differences between subjects. For each subject, we calculated the mean and standard deviation. Subsequently, we subtracted the mean from each feature value and divided it by the standard deviation of the same subject. The labels for each subject were shifted by 3 s because facial pain responses typically exhibit a delay of 2–3 s compared to the stimulus. Additionally, we applied a sliding time window with a duration of 10 s, utilized once per second. This involved combining the FAD and EDA-D from ten seconds prior to predict the pain intensity labels for the next time step. The initial ten seconds of data were excluded due to the absence of prior observations.

### 3.3. Classification, Regression, and Fusion

The EDA-D and FAD were used as features for continuous pain intensity recognition (no pain, low, moderate, and severe) and for modality classification (heat and electrical pain stimuli) using RF, LSTM, and LSTM-SW. In alignment with Werner et al. [14] and Othman et al. [26,41], we trained the Random Forest classifier (RFc) and the Random Forest regression (RFr) with 100 trees and a maximum depth of 10 nodes for classification and regression tasks. Both RFc and RFr were the baseline methods to compare them with LSTM and LSTM-SW methods in this study. Figure 3 shows the six LSTM architectures used in this work: four for classification (A(c), B(c), C(c), and D(c)) and two for regression (A(r) and B(r)). EDA-D or FAD was used as input but not both.

The input size for EDA-D is 10 × 12, and for FAD, it is 10 × 252, where 10 represents timesteps, and 10 × 252 represents features. The classification architectures A(c) and C(c) consist of a single LSTM layer with 4 units activated via ReLU, followed by a flatten layer, and then a dense layer with 128 neurons activated via ReLU. The final dense output layer has 7 neurons in A(c) and 4 neurons in C(c). The classification architectures B(c) and D(c) include a single LSTM layer with 8 units activated via ReLU, followed by a flatten layer, and then a dense layer with 64 neurons activated via ReLU. The final dense output layer has 7 neurons in B(c) and 4 neurons in D(c). The output layer was activated using the Softmax function, and the loss function employed was Categorical Cross-Entropy (CCE). The LSTM with this loss is referred to as LSTM-CCE. The configurations of the regression architecture A(r) are similar to those of A(c) and C(c), and the configurations of the regression architecture B(r) are similar to those of B(c) and D(c), with the exception of the final dense output layer, which has 1 neuron. The output layer was activated using the Sigmoid function, and the loss function used was Binary Cross-Entropy (BCE). The LSTM with this loss is known as LSTM-BCE. The models were trained for 2000 epochs with learning rates of 10${}^{-4}$, 10${}^{-5}$, or 10${}^{-6}$, with a batch size of 512 and using the Adam optimizer. In LSTM-SW, the samples were trained on LSTM after augmenting some training samples using the sample weighting method [25]. RFc with FAD was used to identify samples with prediction scores higher than 0.3 in training datasets, and these samples were duplicated once. LSTM-SW using CCE is referred to as LSTM-SW-CCE, and LSTM-SW using BCE is called LSTM-SW-BCE.

In this work, we applied late fusion, known as Decision Fusion (DF), on the outputs from individually trained models that utilize both EDA and facial expressions modalities. The classification models (RFc, LSTM, and LSTM-SW) yield scores for each potential class, while the regression models (RFr, LSTM, and LSTM-SW) predict continuous values. We aggregated the classifier scores and regression outputs individually into a final decision using a fixed mapping method. Regarding classification, DF was implemented by calculating the mean of output scores per class of both models (one using EDA-D and the other using FAD), then the class with the highest score was selected. Regarding regression, all RFr, LSTM, and LSTM-SW predictions were averaged individually in terms of calculating DF.

## 4. Results

This section presents the results of pain intensity recognition using EDA-D, FAD, and DF. It includes two types of pain stimuli (P = phasic and T = tonic) for each modality variant (H = heat and E = electrical) across three intensities (1 = low, 2 = moderate, and 3 = severe). A seven-class pain intensity recognition was considered as follows, where BL represents no pain: (1) BL, PH1, PH2, PH3, PE1, PE2, and PE3, and (2) BL, TH1, TH2, TH3, TE1, TE2, and TE3. Additionally, four-class pain intensity recognition was considered as follows: (1) BL, PH1, PH2, and PH3; (2) BL, PE1, PE2, and PE3; (3) BL, TH1, TH2, and TH3; and (4) BL, TE1, TE2, and TE3. LSTM and LSTM-SW were compared to the guessing approach (Trivial = majority of the vote, corresponding to no pain labels in the X-ITE Pain Database) and baseline methods (RFc and RFr) across 11 datasets for classification and regression tasks. The results indicate that most LSTM and LSTM-SW models outperformed the Trivial and most RF (RFc and RFr) models. Regarding classification, RFc showed the best performance with small dataset sizes (such as tonic datasets), while DF performed the best across most datasets. In contrast, the regression models demonstrated better performances than classification models on all imbalanced datasets when utilizing EDA, as indicated in the MSE and ICC measures, except with RTD. In the case of the almost balanced dataset (HTD), classification performed the best when using DF. See the sections below for details.

### 4.1. Classification

Figure 4 and Table 2 present the comparison of the performance between RFc, LSTM-CCE, and LSTM-SW-CCE for continuous pain intensity monitoring in terms of the Micro avg. F1-score measure. All RFc, LSTM-CCE, and LSTM-SW-CCE models significantly outperformed the Trivial. Additionally, LSTM-SW-CCE using EDA-D and DF demonstrated similar performances, both significantly outperforming RFc with phasic Subsets (PD, HPD, and EPD) at approximately 50%, 51%, and 66%, respectively. LSTM-CCE and LSTM-SW-CCE using DF with Reduced Subsets (RPD, RHPD, and REPD) improved the performance compared to most EDA and FAD models, with both models performing quite similarly. With HPD, LSTM-SW-CCE using FAD, EDA-D, and DF performed about 16%. Furthermore, RFc using DF with the imbalanced tonic dataset (TD, ETD, RTD, and RETD) performed about 8%, 8%, 23%, and 30%, respectively. With the almost balanced dataset (HTD), LSTM-CCE using DF obtained the highest performance (about 61%). Figure 4 shows how DF improved the performance with the almost balanced dataset (HTD) and the datasets after reducing the imbalanced problem.

### 4.2. Classification vs. Regression

This section presents a comparison among the Trivial, RF, LSTM, and LSTM-SW models concerning both classification and regression. The Mean Squared Error (MSE) and the Intraclass Correlation Coefficient (ICC) [43] were utilized to evaluate the performance of classification models in contrast to regression models. See the following sections: seven-class and four-class pain intensity recognition. Figure 5 shows the results after comparing the performance between classification and regression models.

#### 4.2.1. Heat and Electrical Pain Intensity Recognition (Seven-Class)

In terms of MSE and ICC measures, Table 3 shows the results of classifying all seven available classes using phasic datasets (PD and RPD) and tonic datasets (TD and RTD) while considering various pain stimulus intensities and modalities. All automatic models utilizing EDA-D and DF demonstrated superior performances in both classification and regression compared to models employing FAD. In line with our recent study [27,41], the models with EDA-D and DF outperformed the Trivial and baseline methods (RFr and RFc). DF exhibited a significant enhancement in performance compared to the best FAD models. Further, most EDA-D models outperformed those using DF to recognize continuous pain intensity. The most remarkable outcomes were achieved via (1) LSTM-BCE models using EDA-D with PD and TD datasets, yielding an MSE of 0.06 and 0.08, and an ICC of 0.43 and 0.12, respectively; (2) LSTM-SW-BCE models using EDA-D with RPD dataset, resulting in an MSE of 0.04 and an ICC of 0.84; and (3) LSTM-SW-CCE models using EDA-D with RTD dataset, yielding an MSE of 0.11 and an ICC of 0.31.

#### 4.2.2. Heat Pain Intensity Recognition (Four-Class)

Due to the results from the seven-class pain intensity recognition discussed in the previous section, the four-class models were trained to simplify the problem and enhance performance by focusing on the samples that were exposed to heat stimuli. The HPD and RHPD datasets were obtained by excluding samples related to electrical phasic pain intensities, while the HTD dataset was obtained by excluding samples associated with electrical tonic pain intensities; see Table 3. The top four-class models results are as follows: (1) LSTM-BCE-SW models using DF with HPD, which yielded the highest ICC value of 0.33 along with an MSE of 0.08; furthermore, LSTM-BCE models using EDA-D achieved an ICC of 0.31 and the lowest MSE of 0.07; (2) LSTM-CCE models using EDA and DF with HTD showed good performance with ICC values of 0.33 and 0.35 along with MSE values of 0.15 and 0.16, respectively; and (3) LSTM-BCE models using EDA-D with RHPD achieved the highest ICC value of 0.81, along with the lowest MSE of 0.05.

#### 4.2.3. Electrical Pain Intensity Recognition (Four-Class)

The results of the four-class models trained with electrical phasic pain datasets (EPD and REPD) and electrical tonic pain datasets (ETD and RETD) were presented in Table 3. Samples related to heat pain intensities were excluded from PD, RPD, TD, and RTD. The electrical pain recognition models using EDA-D and DF showed superior performances compared to the Trivial approach and baseline methods (RFr and RFc). LSTM-SW-BCE models using EDA-D with EPD, ETD, and REPD performed the best, achieving the highest ICC values (0.53, 0.21, and 0.88) along with MSE values of 0.05, 0.07, and 0.03, respectively. Additionally, LSTM-BCE models using EDA-D with RETD yielded the highest ICC value (0.49) and the lowest MSE value (0.10).

## 5. Discussion

In this work, we conducted several experiments to compare the performance of three different methods for the automatic monitoring of continuous pain intensity using the X-ITE Pain Database. We used EDA and facial expressions as individual modalities, as well as the fused modality obtained by combining these two modalities. The results in both phasic and tonic datasets show that it is possible to monitor continuous pain intensity; see Figure 4 and Figure 5 and Table 2 and Table 3. The analysis of facial expression features has been previously reported in our recent study [27,41], and we use them here for comparison. For both classification and regression tasks, we trained RF (RFc and RFr), LSTM, and LSTM-SW models using the EDA and facial expression modalities, as well as the late fusion of these modalities (fused modality). We applied a sliding window approach to obtain input samples of 10-s length, and the labels for each subject were shifted by 3 s. All models were trained using 11 datasets derived from the X-ITE Pain Database. This dataset splitting strategy aimed to address the imbalanced problem of the database, improve the results, and ensure the generalizability of the proposed system’s capabilities.

The results indicate that models using the A(c), A(r), and C(c) architectures outperformed those utilizing the B(c), B(r), and D(c) architectures. Furthermore, both LSTM and LSTM-SW models utilizing the EDA modality and fused modality demonstrated significantly superior performance compared to guessing (Trivial). The Trivial approach consistently votes for the majority class (no pain in our experiment). Consistent with the findings of Werner et al. [14,20], the classification LSTM model (LSTM-CCE) using the fused modality with the HTD dataset improved performance compared to individual modalities (EDA and facial expressions), as shown in Table 2 and Table 3.

Furthermore, RFc using DF with EPD performed worse because most of the no pain samples were labeled with pain after the fusion of the FAD and EDA-D modalities. The possible reason behind this result is that RTD is a combination of HTD and RETD datasets; RTD contains less imbalanced data due to the influence of the almost balanced HTD dataset (which accounts for only 20% of samples experiencing no pain, as shown in Table 1).

Fused modality performed the best with the phasic Reduced Subsets (RPD, RHPD, and REPD) compared to using EDA and facial expressions modalities individually. See the results for EDA-D, FAD, and DF (which indicates the fused modality) in Table 2. LSTM-SW-CCE outperformed LSTM-SW when using the EDA modality with phasic Subsets, but the fused modality did not improve the performance. Furthermore, RFc using the fused modality performed the best with the tonic datasets (TD, ETD, RTD, and RETD). The possible reason is that RFc performs well with small data sizes. After comparing classification and regression results, we found that regression was better than classification with most imbalanced datasets (see Figure 5 and Table 3). Additionally, most of the LSTM and LSTM-SW models using the EDA modality achieved the best performance. They outperformed the models that used FAD and the fused modality, except with the HPD dataset. LSTM-SW-BCE using the fused modality with the HPD dataset improved the ICC performance from 0.32 to 0.33 (as shown in Table 3); however, the improvement was not significant. The best performance in terms of the ICC measure was achieved via LSTM-BCE using the EDA modality with the REPD dataset (ICC approximately 0.88, indicating very good performance). This improvement might be due to reducing noise or outlier data and including more data of pain intensities.

LSTM-SW increased the performance compared to LSTM in several models. LSTM-SW-CCE performed the best when using EDA-D with PD, HPD, EPD, and ETD datasets in terms of the Micro avg. F1-score measure (see Table 2), and LSTM-SW-BCE outperformed LSTM-CCE when using EDA-D with EPD, ETD, RPD, and REPD datasets in terms of the ICC measure (see Table 3). This leads to the hypothesis that the success of LSTM-SW is based on downweighting samples in the training set with a lower facial response using the sample weighting method [26]. These samples might negatively affect the model’s performance.

## 6. Conclusions

This study advances the recognition of continuous pain intensity using EDA and facial expressions modalities with the X-ITE Pain Database. Unlike prior works [14,20] that focused on specific time windows from the database, we utilized most of the data from the complete continuous recording phase. Three methods, namely RF (RFc and RFr), LSTM, and LSTM-SW, were applied to individual modalities (EDA and facial expressions) as well as a fused modality (both modalities combined). To address the data imbalance and outliers, the database was split into six datasets based on different qualities of pain stimuli. The performance increased by reducing the noise in EDA and facial expressions data. We removed some no-pain samples prior to pain intensity samples in a time series for each subject in every six datasets due to inconsistencies between plenty no pain labels and the facial expressions responses [27,41]. This study’s findings suggest that for balanced or almost balanced datasets, the classification using the fused modality of EDA and facial expressions is preferable for pain intensity recognition. However, with imbalanced datasets, regression using the EDA modality performs best. RFc and RFr are the best with the small size of datasets; however, the performance is still poor. LSTM and LSTM-SW performed well with big sizes of datasets. This study confirms that it is possible to continuously monitor pain intensity using machine learning models with facial expression and EDA signal only.

Although the results of this work are promising, some limitations have been identified that need to be addressed for the further advancement of this system. These limitations include the fact that the X-ITE Pain Database is based on healthy participants, the relatively small size of the training data, and the requirement for extracting informative features. Several suggested ways to overcome these limitations such as the following: the proposed system should be applied on real patients before it can be considered ready for clinical studies; acquiring a larger dataset with more pain intensities is necessary for more reliable automatic monitoring of continuous pain intensity; more statistical measures of the time series should be used to improve system performance (using the remaining of the statistical measures in the Werner et al. [24] study).

## Figures and Tables

**Figure 1 life-13-01828-f001:**
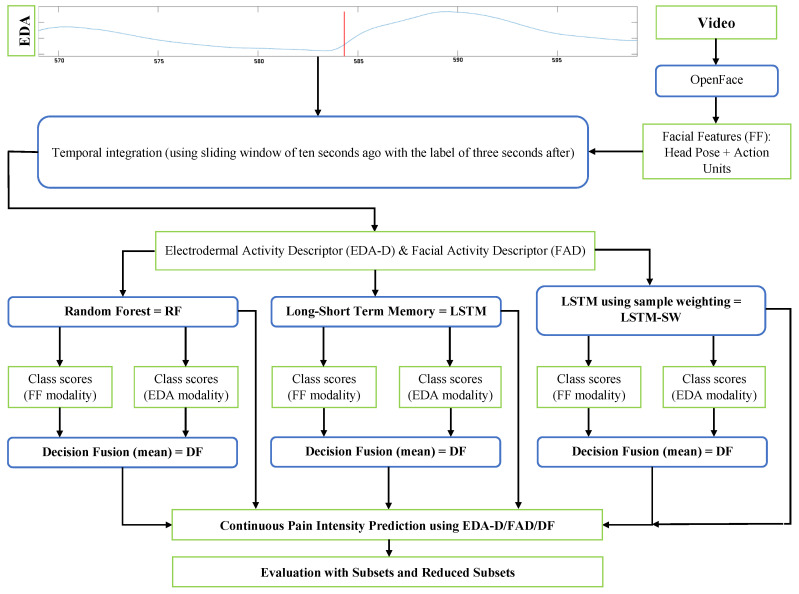
Overview of continuous pain intensity monitoring system.

**Figure 2 life-13-01828-f002:**
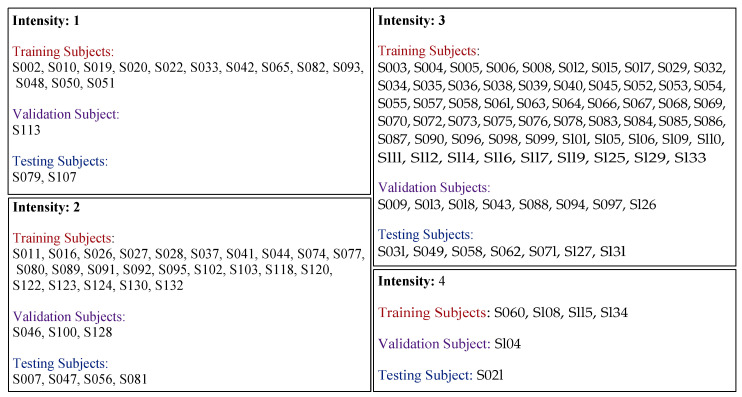
Assignment of subjects to categories of facial response intensities. Intensity 1 = lack of facial responses to pain. Intensity 2, 3 = moderate intensity of facial responses to pain. Intensity 4 = intensive facial responses to pain.

**Figure 3 life-13-01828-f003:**
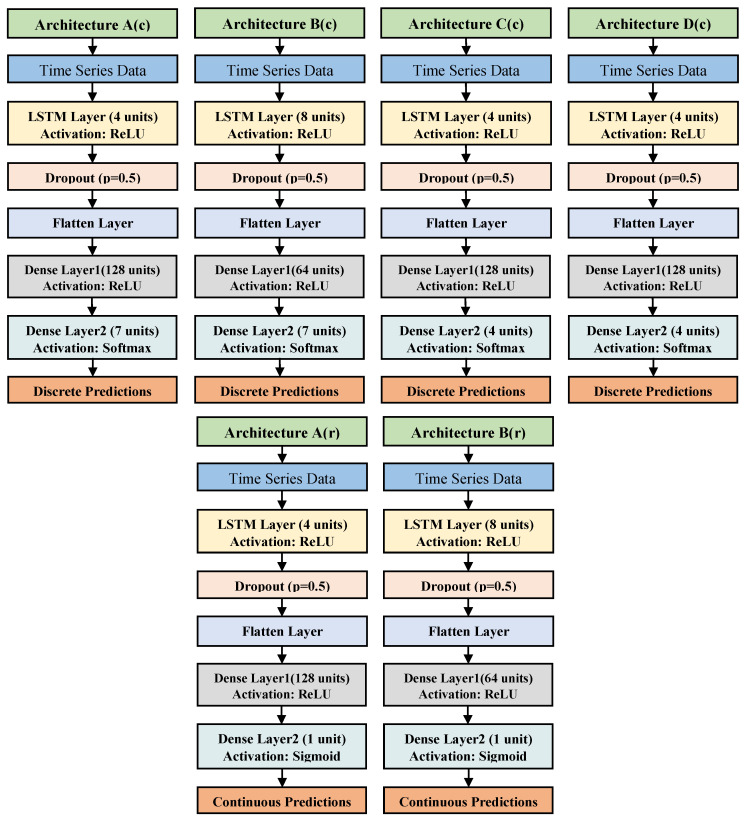
Architectures configurations for classification and regression methods.

**Figure 4 life-13-01828-f004:**
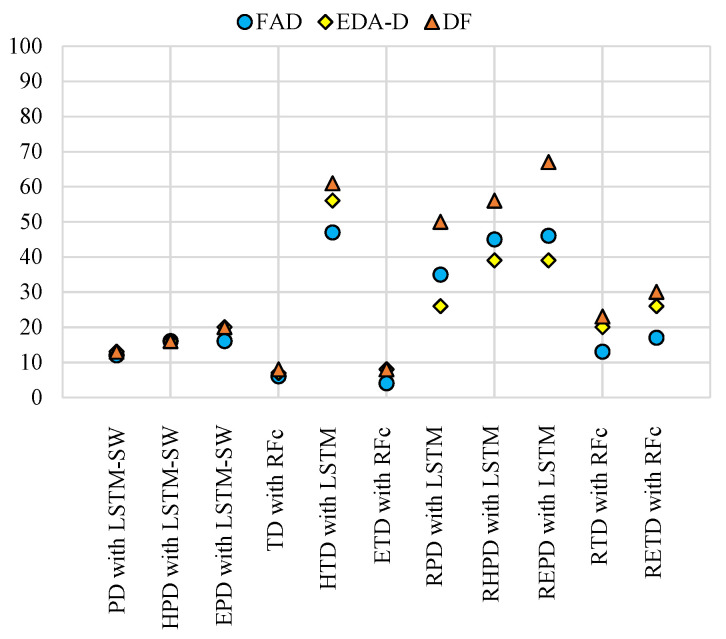
The best results from RFc, LSTM, and LSTM-SW when using FAD, EDA-D, and DF in terms of Micro avg. F1-score (%) measure. FAD: Facial Activity Description, EDA-D: EDA Description, and DF: Decision Fusion. All results are shown in Table 2.

**Figure 5 life-13-01828-f005:**
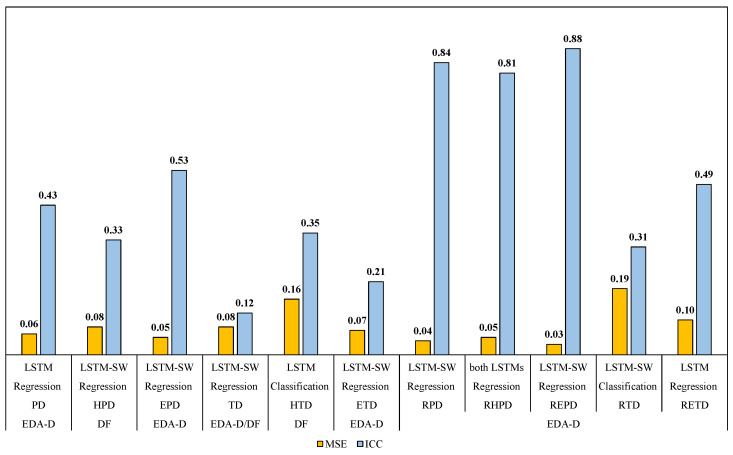
The best results from RFc, LSTM, and LSTM-SW when using FAD, EDA-D, and DF in terms of MSE and ICC measures. EDA-D: EDA Description, and DF: Decision Fusion. All results are shown in Table 3.

**Table 1 life-13-01828-t001:** Samples’ distribution based on labels.

Subsets		No	Pain	Reduced Subsets		No	Pain
		Pain	Intensities			Pain	Intensities
PD	Phasic Dataset	77.7%	22.3%	RPD	Reduced Phasic Dataset	50.0%	50.0%
HPD	Heat Phasic Dataset	78.5%	21.5%	RHPD	Reduced Heat Phasic Dataset	50.1%	49.9%
EPD	Electrical Phasic Dataset	86.1%	13.9%	REPD	Reduced Electrical Phasic Dataset	50.0%	50.0%
TD	Tonic Dataset	70.3%	29.7%	RTD	Reduced Tonic Dataset	38.1%	61.9%
HTD	Heat Tonic Dataset	20.0%	80.0%				
ETD	Electrical Tonic Dataset	82.0%	18.0%	RETD	Reduced Electrical Tonic Dataset	49.0%	51.0%

**Table 2 life-13-01828-t002:** Comparison of Trivial, baseline (RFc), LSTM-CCE and LSTM-SW-CCE models in terms of Micro avg. F1-score measure (%). The best results were highlighted with cyan (FAD), yellow (EDA-D), and orange (DF) colors. Trivial: guessing (majority of vote = no pain), FAD: Facial Activity Description, EDA-D: EDA Description, DF: Decision Fusion, and CCE: Categorical Cross-Entropy loss.

		Models								
Datasets		n-Class	Trivial	RFc						
				FAD		EDA-D		DF		
Subsets	PD	7	0	6		9		7		
	HPD	4	0	10		12		4		
	EPD	4	0	10		15		13		
	TD	7	0	6		7		8		
	HTD	4	0	38		46		53		
	ETD	4	0	4		8		8		
Reduced Subsets	RPD	7	0	15		20		24		
	RHPD	4	0	24		28		30		
	REPD	4	0	28		36		44		
	RTD	7	0	13		20		23		
	RETD	4	0	17		26		30		
		Models								
Datasets		n-Class	Archit.	LSTM-CCE			LSTM-SW-CCE			lr
				FAD	EDA-D	DF	FAD	EDA-D	DF	
Subsets	PD	7	A(c)	10	10	8	12	13	13	10${}^{-5}$
	HPD	4	C(c)	16	13	13	16	16	16	
	EPD	4	C(c)	15	16	14	16	20	20	
	TD	7	A(c)	3	3	0.5	6	7	4	10${}^{-6}$
	HTD	4	C(c)	47	56	61	48	56	58	
	ETD	4	C(c)	6	4	4	6	3	0.7	
Reduced Subsets	RPD	7	A(c)	35	26	50	32	29	50	10${}^{-4}$
	RHPD	4	C(c)	45	39	52	44	37	51	
	REPD	4	C(c)	46	39	67	46	45	66	
	RTD	7	B(c)	8	15	11	12	14	15	10${}^{-6}$
	RETD	4	D(c)	12	22	20	11	22	21	

**Table 3 life-13-01828-t003:** Comparison of Trivial, baseline (RFc), LSTM-CCE, LSTM-SW-CCE, LSTM-BCE, and LSTM-SW-BCE models with MSE and ICC measures. The top results were highlighted with cyan (FAD), yellow (EDA-D), and orange (DF) colors. FAD: Facial Activity Description, EDA-D: EDA Description, DF: Decision Fusion, CCE: Categorical Cross-Entropy loss, Meas.:Measurements, Trivial: guessing (majority of vote = no pain), Red. Subset: Reduced Subset, Archit.: Architecture, and lr: learning rate.

	Task				Classification									Regression										lr
	Models			RFc			Archit.	LSTM-CCE			LSTM-SW-CCE			RFr			Archit.	LSTM-BCE			LSTM-SW-BCE			
Meas.			Trivial	FAD	EDA-D	DF		FAD	EDA-D	DF	FAD	EDA-D	DF	FAD	EDA-D	DF		FAD	EDA-D	DF	FAD	EDA-D	DF	
MSE	Subsets	PD	0.10	0.10	0.09	0.10	A(c)	0.09	0.09	0.09	0.10	0.09	0.09	0.09	0.07	0.07	A(r)	0.08	0.06	0.06	0.08	0.08	0.06	10${}^{-5}$
		HPD	0.11	0.11	0.11	0.11	C(c)	0.11	0.10	0.10	0.11	0.11	0.10	0.09	0.09	0.08	C(r)	0.08	0.08	0.07	0.09	0.08	0.08	
		EPD	0.07	0.07	0.06	0.45	C(c)	0.07	0.06	0.06	0.08	0.06	0.06	0.06	0.05	0.05	C(r)	0.05	0.04	0.04	0.06	0.05	0.04	
		TD	0.12	0.12	0.16	0.12	A(c)	0.12	0.12	0.12	0.12	0.12	0.12	0.10	0.13	0.10	A(r)	0.09	0.08	0.08	0.09	0.09	0.09	10${}^{-6}$
		HTD	0.41	0.25	0.18	0.18	C(c)	0.18	0.15	0.16	0.16	0.15	0.16	0.13	0.13	0.11	C(r)	0.12	0.11	0.11	0.16	0.11	0.12	
		ETD	0.09	0.09	0.11	0.09	C(c)	0.09	0.08	0.08	0.08	0.09	0.10	0.09	0.09	0.08	C(r)	0.07	0.07	0.07	0.08	0.07	0.07	
	Red. Subsets	RPD	0.23	0.20	0.14	0.16	A(c)	0.12	0.05	0.07	0.14	0.05	0.07	0.12	0.09	0.10	A(r)	0.08	0.04	0.05	0.09	0.04	0.04	10${}^{-4}$
		RHPD	0.26	0.23	0.22	0.20	C(c)	0.13	0.08	0.09	0.14	0.07	0.09	0.13	0.13	0.12	C(r)	0.09	0.05	0.05	0.09	0.05	0.05	
		REPD	0.26	0.21	0.12	0.13	C(c)	0.13	0.05	0.06	0.15	0.05	0.06	0.12	0.08	0.08	C(r)	0.10	0.04	0.05	0.11	0.03	0.05	
		RTD	0.25	0.23	0.18	0.20	B(c)	0.25	0.21	0.23	0.24	0.19	0.22	0.14	0.13	0.12	B(r)	0.13	0.11	0.11	0.13	0.11	0.11	10${}^{-6}$
		RETD	0.25	0.24	0.18	0.19	D(c)	0.24	0.16	0.19	0.26	0.16	0.20	0.15	0.12	0.12	D(r)	0.14	0.10	0.10	0.13	0.10	0.10	
ICC	Subsets	PD	0	0.10	0.33	0.16	A(c)	0.18	0.30	0.17	0.20	0.40	0.24	0.13	0.41	0.30	A(r)	0.20	0.43	0.34	0.22	0.20	0.37	10${}^{-5}$
		HPD	0	0.16	0.11	0.07	C(c)	0.26	0.30	0.25	0.26	0.29	0.30	0.19	0.20	0.21	C(r)	0.28	0.28	0.31	0.27	0.32	0.33	
		EPD	0	0.16	0.37	0.16	C(c)	0.25	0.36	0.26	0.24	0.50	0.33	0.18	0.47	0.37	C(r)	0.27	0.49	0.42	0.28	0.53	0.47	
		TD	0	0.08	0.10	0.11	A(c)	0.07	0.07	0.01	0.08	0.11	0.07	0.10	0.09	0.10	A(r)	0.11	0.12	0.12	0.12	0.11	0.11	10${}^{-6}$
		HTD	0	0.11	0.30	0.30	C(c)	0.19	0.33	0.35	0.20	0.31	0.29	0.17	0.31	0.27	C(r)	0.15	0.28	0.23	0.15	0.30	0.24	
		ETD	0	0.07	0.14	0.12	C(c)	0.09	0.09	0.06	0.11	0.00	0.01	0.09	0.17	0.14	C(r)	0.09	0.09	0.09	0.11	0.21	0.17	
	Red. Subsets	RPD	0	0.19	0.44	0.40	A(c)	0.57	0.83	0.76	0.49	0.83	0.76	0.23	0.45	0.37	A(r)	0.56	0.81	0.75	0.54	0.84	0.77	10${}^{-4}$
		RHPD	0	0.21	0.23	0.31	C(c)	0.56	0.75	0.71	0.55	0.76	0.71	0.26	0.24	0.27	C(r)	0.62	0.81	0.77	0.62	0.81	0.76	
		REPD	0	0.27	0.58	0.56	C(c)	0.56	0.84	0.81	0.52	0.84	0.80	0.32	0.63	0.53	C(r)	0.52	0.86	0.79	0.51	0.88	0.79	
		RTD	0	0.09	0.21	0.20	B(c)	0.05	0.25	0.16	0.08	0.31	0.21	0.05	0.18	0.12	B(r)	0.04	0.23	0.14	0.07	0.24	0.16	10${}^{-6}$
		RETD	0	0.12	0.37	0.32	D(c)	0.15	0.47	0.34	0.10	0.46	0.31	0.15	0.34	0.27	D(r)	0.09	0.49	0.33	0.20	0.44	0.35	

## Data Availability

Data are available from the authors upon request (to Sascha Gruss or Steffen Walter) for researchers of academic institutes who meet the criteria for access to the confidential data.

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
