# Peer review of "Automated Electrodermal Activity and Facial Expression Analysis for Continuous Pain Intensity Monitoring on the X-ITE Pain Database"

_life, 2023, doi:10.3390/life13091828_

Round 1

Reviewer 1 Report

Dear authors,

I read with great interest the manuscript, which falls within the aim of this Journal. In my honest

opinion, the topic is interesting enough to attract the readers’ attention. Nevertheless, authors should clarify some points and improve the discussion, as suggested below. Authors should consider the

following recommendations:

In my opinion you have to refer to updated literature and how this machine learing can be applied as in infertility in the pain management by endometriosis.

I suggest you to read and cite these articles:

The Future Is Coming: Artificial Intelligence in the Treatment of Infertility Could Improve Assisted Reproduction Outcomes—The Value of Regulatory Frameworks

Impact of lifestyle and diet on endometriosis: a fresh look to a busy corner

 Minor editing of English language required

Author Response

Reviewer 1
In my opinion you have to refer to updated literature and how this machine learing can be applied as in infertility in the pain management by endometriosis.
I suggest you read and cite these articles:

The Future Is Coming: Artificial Intelligence in the Treatment of Infertility Could Improve Assisted
Reproduction Outcomes—The Value of Regulatory Frameworks
Impact of lifestyle and diet on endometriosis: a fresh look to a busy corner
Response 1: Thank you for this suggestion. We have read both suggested articles, it seems there is no correlation between them and our paper. In this paper, we focus on introducing an automated system for continuous pain intensity monitoring using Electrodermal Activity (EDA) and facial expressions, employing different machine learning techniques, while the first article explores the potential applications of artificial intelligence (AI) in improving infertility diagnosis and treatment outcomes, particularly in assisted reproduction technologies (ART), with a focus on clinical AI applications and ethical considerations; and the second suggested article addresses the chronic inflammatory disorder of endometriosis, exploring potential interventions such as fish oil and vitamins to alleviate symptoms and the need for further research in this area.

Reviewer 2 Report

This work reports the development of an automated system that continuously monitors patient pain intensity. The document includes valuable material for publication and follows the journal's scope. The work contains helpful information for the public of this magazine. The methodological strategy followed in the research is consistent with the outcome. In addition, the results are fully presented and clearly discussed. Some aspects need to be improved before being published. Here are the main observations:

1. Using adjectives such as "best" are not usual in scientific texts since they lack objectivity. This word appears 29 times in the text. I suggest reviewing the wording of the manuscript in detail and avoiding as far as possible adjectives of this type; failing that, I suggest that they establish a reference to indicate "because x thing is better than y thing" (e.g., page 3, line 106).

2.     The document contains some grammar and writing errors.

3.     The conclusion seems to be very long. Conclusions written with short sentences that respond to the proposed objectives are preferred since they provide specific information to the reader and highlight the highlights of the work.

The document contains some grammar and writing mistakes. Please thoroughly review the wording of the whole document.

Author Response

Reviewer 2
This work reports the development of an automated system that continuously monitors patient pain intensity. The document includes valuable material for publication and follows the journal's scope. The work contains helpful information for the public of this magazine. The methodological strategy followed in the research is consistent with the outcome. In addition, the results are fully presented and clearly discussed. Some aspects need to be improved before being published. Here are the main observations:

Point 1: Using adjectives such as "best" are not usual in scientific texts since they lack objectivity. This word appears 29 times in the text. I suggest reviewing the wording of the manuscript in detail and avoiding as far as possible adjectives of this type; failing that, I suggest that they establish a reference to indicate "because x thing is better than y thing" (e.g., page 3, line 106).
Response 1: Thank you for this suggestion. We have avoided using best in whole manuscript, they were replaced.

Point 2: The document contains some grammar and writing errors.
Response 2: Thanks, the manuscript has been reviewed and the errors has been corrected

Point 3: The conclusion seems to be very long. Conclusions written with short sentences that respond to the proposed objectives are preferred since they provide specific information to the reader and highlight the highlights of the work.
Response 3: Thank you for this suggestion. We have rephrased and summarized the conclusion section, and we added limitations and future work paragraph.

Author Response

Reviewer 3
In this manuscript, the authors proposed a method for continuous automated monitoring of patient pain by analyzing electrodermal activity and facial expressions using the X-ITE pain database. Although this work is practical and logical, some major concerns are raised to be addressed properly as follows.

Point 1: The current manuscript needs to be polished by a native English speaker or professional language editing service.
Response 1: Thank you for this suggestion, the manuscript has been reviewed and the errors has been corrected.

Point 2: To enhance the abstract's readability, authors are encouraged to employ succinct sentence structures and minimize the use of excessive technical terminology. Employing clear and concise language will aid readers in comprehending the text's essence more effectively.
Response 2: Thank you for this suggestion. We have rephrased and summarized the abstract section.

Point 3: Although the statistics amply support the conclusions reached, we think they could be presented in a more enticing and compelling way to more effectively communicate their relevance and influence.
Response 3: Thank you for this suggestion. We have rephrased and summarized the conclusion section, and we added limitations and future work paragraph.

Point 4: The conclusion section, which currently appears somewhat as an afterthought, could benefit from expansion. Authors are advised to underscore noteworthy
Response 4: Thank you for this suggestion. We have rephrased and summarized the conclusion section, and we added limitations and future work paragraph.

Reviewer 4 Report

Title

The title of the paper is well-chosen, accurately reflecting the main focus of the research. However, a slight adjustment could enhance its specificity. Consider revising the wording “Facial Expression Analysis” to “Automated Facial Expression Analysis.”

Abstract

  • Grammar and Syntax: The abstract has minor grammatical issues that need correction. The phrase “continuously monitor patient pain intensity” should be changed to “continuously monitors patient pain intensity,” and “analysing” should be replaced with “analyzes.” Also, correct “play a a prominent role” to “play a prominent role.”
  • Clarity: The introduction to the modalities could be clearer by saying, "The EDA sensor modality and facial expression analysis, two highly informative modalities...".
  • Novelty: Be explicit about the uniqueness of the study by specifying how your approach or methods differ from previous studies and what new insights they offer.
  • Results: The results section of the abstract is somewhat dense; simplify it by focusing on the most significant findings and adding information about the potential practical implications.

Keywords

The keywords are well chosen, but consider eliminating redundancy, such as removing "modalities."

Introduction

  • Repetitiveness: Avoid repetition, such as the twice-appearing sentence "both modalities are good measures for pain assessment [6,7]."
  • Grammar and Syntax: Be mindful of grammar; for instance, change "An reliable assessment of pain" to "A reliable assessment of pain." Also, maintain tense consistency.

1. Related Work

  • Typographical Errors and Phrasing: Address minor typographical errors and awkward phrasings, such as "analysed the spectral of EDA" and clarify sentences like "RFc and RFr as baseline methods to continuously monitor pain intensity with the X-ITE Pain Database."
  • Differentiation: Emphasize what sets this study apart from others more clearly.
  • Consistent Detail Level: Maintain a uniform level of detail, providing enough context without overwhelming specifics.
  • Critical Analysis: Include strengths and weaknesses of previous approaches, addressing how this study fills gaps in the literature.

2. Materials and Methods

  • Definitions: Ensure all acronyms and specific terms are defined before use.
  • Complex Sentences: Break down complex sentences into smaller, clearer statements.

Results

  • Summary: Provide a brief overview of the main findings at the beginning.
  • Formatting: Correct inconsistencies like "%30."
  • Clarity in Comparison: Summarize dense comparisons between models and measures.
  • Proofreading: Carefully proofread for errors, such as an unexpected semicolon in "FAD, EDA-D, and; DF performed the best..."

Conclusions

  • Limitations and Future Research: Address method or data limitations, and suggest future research directions.
  • Concluding Statement: End with a robust statement emphasizing the importance of the work.

General Remarks

The paper provides valuable insights into pain intensity monitoring and offers a detailed analysis. Addressing the above points would enhance clarity, readability, and overall impact. The practical applications of this research have the potential to contribute significantly to the field, and articulating this potential will provide a strong closing argument for the paper's importance.

Please refer to the comments above.

Author Response

Reviewer 4
Point 1: Title
The title of the paper is well-chosen, accurately reflecting the main focus of the research. However, a slight adjustment could enhance its specificity. Consider revising the wording “Facial Expression Analysis” to “Automated Facial Expression Analysis.”Response 1: Thank you for this suggestion. We have adjusted the title.
Point 2: Abstract

• Grammar and Syntax: The abstract has minor grammatical issues that need correction. The phrase “continuously monitor patient pain intensity” should be changed to “continuously monitors patient pain intensity,” and “analysing” should be replaced with “analyzes.” Also, correct “play a a prominent role” to “play a prominent role.”
• Clarity: The introduction to the modalities could be clearer by saying, "The EDA sensor modality and facial expression analysis, two highly informative modalities...".
• Novelty: Be explicit about the uniqueness of the study by specifying how your approach or methods differ from previous studies and what new insights they offer.
• Results: The results section of the abstract is somewhat dense; simplify it by focusing on the most significant findings and adding information about the potential practical implications.
Keywords
The keywords are well chosen, but consider eliminating redundancy, such as removing "modalities."
Response 2: Thanks. We have rephrased and summarized the abstract section and corrected the suggested points.

Point 3: Introduction
• Repetitiveness: Avoid repetition, such as the twice-appearing sentence "both modalities are good measures for pain assessment [6,7]."
• Grammar and Syntax: Be mindful of grammar; for instance, change "An reliable assessment of pain" to "A reliable assessment of pain." Also, maintain tense consistency.
Response 3: Thank you for this suggestion, the manuscript has been reviewed and the errors have been corrected.

Point 4: Related Work
• Typographical Errors and Phrasing: Address minor typographical errors and awkward phrasings, such as "analysed the spectral of EDA" and clarify sentences like "RFc and RFr as baseline methods to continuously monitor pain intensity with the X-ITE Pain Database."
• Differentiation: Emphasize what sets this study apart from others more clearly.
• Consistent Detail Level: Maintain a uniform level of detail, providing enough context without overwhelming specifics.
• Critical Analysis: Include strengths and weaknesses of previous approaches, addressing how this study fills gaps in the literature.

Response 4: Thank you for this suggestion. We have rephrased the related work section, and we considered all suggested comments.

Point 5: Materials and Methods
• Definitions: Ensure all acronyms and specific terms are defined before use.
• Complex Sentences: Break down complex sentences into smaller, clearer statements.Response 5: Thanks. We have reviewed the acronyms, and we Break down complex sentences into smaller.

Point 6: Results
• Summary: Provide a brief overview of the main findings at the beginning.
This was existed, see page 10 line 382
• Formatting: Correct inconsistencies like "%30."
• Clarity in Comparison: Summarize dense comparisons between models and measures.
• Proofreading: Carefully proofread for errors, such as an unexpected semicolon in "FAD, EDA-D,
and; DF performed the best..."

Response 6: Thanks. The manuscript has been reviewed and the errors have been corrected.

Point 7: Conclusions
• Limitations and Future Research: Address method or data limitations, and suggest future
research directions.
• Concluding Statement: End with a robust statement emphasizing the importance of the work. General Remarks
Response 7: Thank you for this suggestion. We have rephrased and summarized the conclusion section, and we added limitations and future work paragraph.

Round 2

Reviewer 3 Report

The authors have made detailed revisions accordingly.